# Hypergraph Propagation and Community Selection for Objects Retrieval

**Guoyuan An**[1]**, Yuchi Huo**[2,3]**, and Sung-Eui Yoon**[1]

[1]School of Computing, KAIST
[2]Zhejiang Lab
[3] State Key Lab of CAD&CG, Zhejiang University

## Abstract

Spatial verification is a crucial technique for particular object retrieval. It utilizes spatial information for the accurate detection of true positive images. However, existing query expansion and diffusion methods cannot efficiently propagate the spatial information in an ordinary graph with scalar edge weights, resulting in low recall or precision. To tackle these problems, we propose a novel hypergraph-based framework that efficiently propagates spatial information in query time and retrieves an object in the database accurately. Additionally, we propose using the image graph's structure information through community selection technique, to measure the accuracy of the initial search result and to provide correct starting points for hypergraph propagation without heavy spatial verification computations. Experiment results on ROxford and RParis show that our method significantly outperforms the existing query expansion and diffusion methods.

## 1   Introduction

Image search is a fundamental problem in computer vision with numerous applications such as content-based image browsing [36, 19, 1], visual localization [2, 30], and 3D reconstruction [15, 31]. To retrieve the database images that contain the same object of a query, a search engine represents images using global or local features [5, 21, 33, 26, 3] that describe their similarities, and then performs a nearest neighbor search for the query image.

An object is depicted in various conditions in the real world [25]. Global features of images depicting the same object form a manifold in the feature space [39]. Euclidean-distance-based nearest neighbor search [20, 4] does not work well on this manifold. To improve the retrieval performance, query expansion [9, 13, 27, 3, 14] issues a new query by aggregating useful information of nearest neighbors of the prior query. Unfortunately, this approach only explores the neighborhood of very similar images, resulting in a low recall [16]. Diffusion [16, 39] explores more images on a neighborhood graph of the dataset. It, however, adopts false positive items in practice, leading to a low precision [25].

Spatial verification [23] uses spatial information to detect the true positive images. However, due to its heavy computational overhead, it is impractical to apply spatial verification to all the images during diffusion. This paper aims to pre-compute the spatial matching information of database images offline and quickly propagate them for a given query online. To achieve this goal, we propose a hypergraph propagation model. Our model efficiently builds a hypergraph for a given query and propagates the spatial relationship with the query through hyperedges in a sequence of images.

When the query image is not in the database, diffusion represents it using its top-ranked images in the initial search [16, 37]. However, not all the top-ranked images contain the object of the query [16, 25]. Some works [25, 6] show that applying spatial verification before diffusion improves the retrieval

performance, but it requires heavy computational overheads [25, 9]. The second contribution of this paper is the proposal of an initial search technique, community selection, which reduces the overhead of spatial verification while maintaining high accuracy.

Experimental result shows that our hypergraph propagation significantly outperforms the existing query expansion and diffusion methods. On the hard protocol of the ROxford dataset, our hypergraph-propagation-based approach achieves the impressive mAPs of 73.0 and 60.5 with and without R1M distractors, respectively.

In conclusion, our contributions are: 1) we propose a novel hypergraph model to efficiently propagate the spatial information in a sequence of images to improve the search performance; 2) we propose community selection to predict the accuracy of the initial search and to provide correct starting points for hypergraph diffusion; 3) we achieve state-of-the-art performance on the ROxford dataset with and without R1M distractors.

## 2 Background

### 2.1 Query expansion and diffusion: relation based search

Query expansion (QE) [9] aggregates the top-ranked initial candidates into an expanded query, which is then used to search more images in the database. The critical factor here is how to aggregate only helpful information and cull out unrelated information.

Average query expansion (AQE) [9] mean-aggregates the top k retrieved images as the expanded query. Average query expansion with decay (AQEwD) [13] gives the top k images the monotonically decaying weights over their rank. Alpha query expansion ($\alpha$QE) [27] uses the power-normalized similarity between the query and the top-ranked images as the aggregation weights. Discriminative query expansion (DQE) [3] also uses the weighted average, where the weight is the dual-form solution of an SVM, which classifies the positive (top-ranked) and negative (low-ranked) images of a query. Learnable attention-based query expansion (LAttQE) [14] uses self-attention [34] to share information between the query and the top-ranked items to compute better aggregation weights.

Unfortunately, these QE methods only explore the neighborhood of very similar images of the query. The recursive methods [9] improve the results by recursively generating new queries from all spatially verified results returned so far. However, this is at the cost of increased query time.

Diffusion [39, 16] ameliorates the problem mentioned above by propagating similarities through a pairwise affinity matrix. The works of Chang *et al.* [6] and Liu *et al.* [17] are two novel variants of this line. The critical issue in diffusion is how to calculate the affinity matrix. Some works [24, 16] use the reciprocal neighborhood relations to refine the search results, and other [23, 6] use the inliers number of SP to re-weight the similarity. However, all of these methods compress the relations between two images as a single scala. As a result, these prior methods lost the spatial matching detail. Although the propagation methods are elegant, they have difficulties recovering good relations to guide the diffusion direction.

We propose a novel and straightforward hypergraph model to utilize the relations among database images better. It connects the corresponding local features among images using hyperedges; in a sequence of similar images, it detects the image from which the following images are no longer relevant to the query.

### 2.2 Local descriptors and spatial verification

Local descriptors are the representations of an image's important patches. Some popular local descriptors are RootSIFT [3], deep local feature (DELF) [21] and deep local and global feature (DELG) [5].

Local descriptors' location information is usually used in the spatial verification (SP) [22] stage. A standard SP pipeline is first using a KD-tree [22] to find the nearest neighbors between local descriptors in two images. Then it uses the RANSAC algorithm [12] to estimate the homography matrix [32] and the number of inlier correspondences between two images. Existing image search engines commonly apply SP on the shortlist, say the top 100 images ranked by a global descriptor.

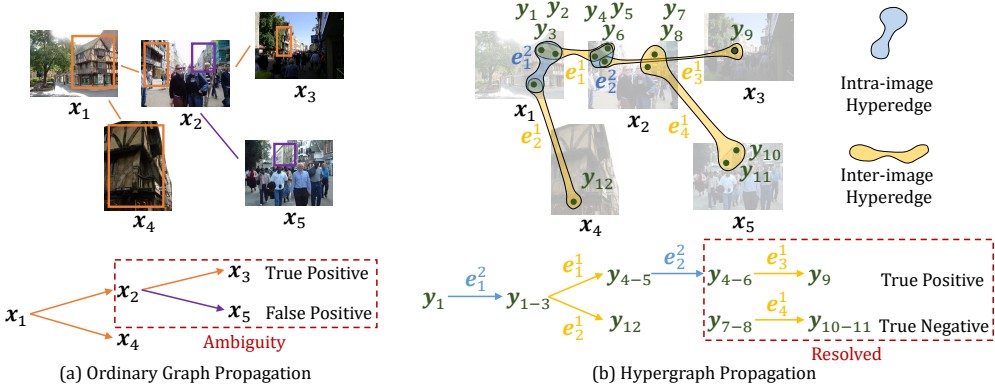

Figure 1:  a) shows a part of an ordinary graph with scalar-weighted, i.e., similarity, edges. Orange frames are the common visible regions among images $\mathbf{x}_1$, $\mathbf{x}_2$, $\mathbf{x}_3$, and $\mathbf{x}_4$. Purple frames are the common visible regions between images $\mathbf{x}_2$ and $\mathbf{x}_5$. $\mathbf{x}_3$ and $\mathbf{x}_5$ are close neighbors to image $\mathbf{x}_2$. While $\mathbf{x}_3$ is related to $\mathbf{x}_1$ by sharing the orange frame, $\mathbf{x}_5$ is not. Utilizing scalar-weighted edges cannot propagate the query in the ordinary graph without this ambiguity issue. b) shows the corresponding hypergraph of a). Inter-image hyperedges $\mathbf{e}_s^1$ are shown in yellow, intra-image hyperedges $\mathbf{e}_k^2$ are in blue, and local features $\mathbf{y}_n$ are in green. A hypergraph path connects local features from $\mathbf{y}_1$ to $\mathbf{y}_9$ in $\mathbf{x}_1$ and $\mathbf{x}_3$, but no path connects local features in $\mathbf{x}_1$ and $\mathbf{x}_5$. A large version of this figure is in the appendix.

SP is known to be crucial for verifying true positive images [25]. Performing SP before diffusion improves the final retrieval performance [25, 6]. However, it is also the slowest process [3, 23] in image search. In this paper, we propose community selection to reduce the computing overhead of SP. Community selection frames the diffusion initialization task as selecting a community for the given query rather than verifying the top-ranked images one by one. Experiment shows that community selection reduces much SP overhead without hurting the accuracy.

## 3   Overview

We first discuss issues of query expansions and diffusion process of ordinary graphs in Sec.3.1 and our novel hypergraph based propagation (Sec. 3.2) and initialization scheme utilizing the concept of community (Sec. 3.3) for achieving accurate diffusion process.

### 3.1   Propagating in ordinary graph

We can use a graph $\boldsymbol{G} = (\boldsymbol{X}, \boldsymbol{F})$ to represent the database images. $\boldsymbol{X} := \{\mathbf{x}_1, ..., \mathbf{x}_m\}$ is the node set, where item $\mathbf{x}_i$ could be either an image or an image region depending on a chosen retrieval method; $\boldsymbol{F}$ is the set of edges whose scalar weight represents the similarity between two nodes connected by each edge. Utilizing information in this ordinary graph is an important strategy to improve retrieval results, and, thus, widely used in various methods, such as query expansion [9, 8, 27, 14] and diffusion [16, 39]. Both diffusion and query expansion can be represented by the following function:

$$\mathbf{f}^* = I(\mathbf{f}^0), \tag{1}$$

where $I$ represents the post-processing process, $f_u \in \mathbb{R}$ is the ranking score for $\mathbf{x}_u$, $\mathbf{f}^0$ is the initial ranking vector with $f_u^0 = 1$ if $\mathbf{x}_u$ is a query and $f_u^0 = 0$ otherwise, and $\mathbf{f}^*$ is the ranking vector after query expansion or diffusion. Diffusion methods diffuse the label of the query node to its neighbors along the graph edges until reaching a stationary state. The stationary state is found by repeating the iteration function: $\mathbf{f}^{t+1} = \alpha \mathbf{O} \mathbf{f}^t + (1 - \alpha)\mathbf{f}^0$, where $\mathbf{O}$ is the normalized affinity matrix that describes the weights of graph edges, $\alpha$ is a jumping probability. Query expansion gathers useful information in the query node's nearest neighbors to update the ranking list.

Diffusion and query expansion have different advantages and disadvantages. Diffusion can efficiently propagate information to the whole graph through a compact formulation. However, the propagation is

conducted through the affinity matrix $\mathbf{O}$ [39, 16], whose scalar value cannot represent accurate spatial information, resulting in low precision. Take Figure 1 as an example, where images of $\mathbf{x}_1$ and $\mathbf{x}_5$ are incorrectly connected through image $\mathbf{x}_2$ in the ordinary graph with scalar edge weights; note that they do not share common regions. This leads to false positives in the diffusion process. We denote this as the **ambiguity problem of propagation**. In contrast, query expansion uses sophisticated approaches, such as transformer [14] and spatial verification [9, 3], to guarantee that only shared information between query and its nearest neighbors are used to update the ranking list. This approach, however, only explores the neighborhood of very similar images due to the computational overhead [25, 3, 9], resulting in a low recall.

This paper proposes novel hypergraph propagation and community selection, settling the ambiguity problem separately in the propagation and initialization stages. We describe these two techniques next.

### 3.2 Extension to hypergraph

One meaningful direction to overcome the drawbacks of diffusion and query expansion is to handle the ambiguity problem of graph propagation by explicitly resolving the propagation in a more informative space. The local features contain informative spatial information that can help to distinguish the ambiguity through geometric verification [22]. However, considering the large scale of the image database, it is impractical to do the geometric verification on all the pairs. In practice, image search engines usually build the kNN graph instead of the complete graph, which means every image only connects with its top k similar images [16]. To utilize the spatial information on the kNN graph, we propose a novel hypergraph propagation model to efficiently propagate spatial information through hyperedges, connecting an arbitrary number of nodes. This technique is appropriate for the real-world situation where the same object in different images has a different number of local features. In the example of Figure 1, image $\mathbf{x}_1$ can connect image $\mathbf{x}_3$ by matching the local features with those of image $\mathbf{x}_2$, but will not connect image $\mathbf{x}_5$.

When a query is given, we define a hypergraph $\boldsymbol{H} = (\boldsymbol{Y}, \boldsymbol{E})$ on top of the ordinary image graph, where its node set $\boldsymbol{Y} = \{\mathbf{y}_1, ..., \mathbf{y}_n\}$ contains all the local features extracted, and the edge set $\boldsymbol{E}$ consists of hyperedges, which are used to connect the local features related to the query object in different images. Our model then propagates the label of local features of the query to their neighbors and uses an aggregation function to find the final ranking list of images, as follows:

$$\mathbf{l}^* = I_h(\mathbf{l}^0), \mathbf{f}^* = Agg(\mathbf{l}^*), \tag{2}$$

where $\mathbf{l}^0$ and $\mathbf{l}^*$ are the initial and final ranking lists of local features respectively, $\mathbf{f}^*$ is the ranking list of images, $I_h$ is the propagation function in a hypergraph, and $Agg$ is the aggregation function that converts ranking scores of local features to ranking scores of images. We will discuss the construction of hypergraph in Section 4.1 and the propagation process in Section 4.2.

### 3.3 Handling new queries: community selection

Diffusion methods [11, 38] usually consider the query point to be contained in the dataset. The standard approach to handle a new query is representing the query using its top-ranked images in the initial search to start the propagation [16]. However, it is hard to guarantee that these global descriptor-based nearest neighbors contain the same object as the query. We call it the **ambiguity problem of initialization**. Recent studies [25, 6] perform online spatial verification on 100 nearest neighbors of the query and delete the wrong neighbors to improve the quality of diffusion. Unfortunately, conducting spatial verification for 100 images in query time is a heavy computational load for a search engine.

We find that the structure of the image graph [23, 3] contains helpful information to measure the accuracy of the nearest neighbors. Following the term in graph theory, we call tightly connected groups communities, which are sets of nodes with many connections inside and few to outside. We observe that the items containing the same objects have a high probability of composing a community and having short geodesic distances. Based on this observation, we frame the diffusion initialization task as selecting a community for the given query instead of using spatial verification to delete the wrong neighbors. This graph-based approach improves both accuracy and speed. We will give the detail of community selection in Section 4.3

# 4 Methods

## 4.1 Construction

Assume that we have an image dataset $X = \{\mathbf{x}_1, ..., \mathbf{x}_m\}$, where $\mathbf{x}_u$ is an image, and its corresponding K-nearest neighbor (Knn) graph [39, 10] $G = (X, F)$, where $F$ is the ordinary edge sets. For hypergraph propagation, we define a hypergraph $H = (Y, E)$, where $Y = \{\mathbf{y}_1, ..., \mathbf{y}_n\}$ is the set of vertices and $\mathbf{y}_i$ represents a local feature in an image. $E = E^1 \cup E^2$ is the union of the inter-image hyperedges set $E^1 = \{\mathbf{e}_1^1, \mathbf{e}_2^1, ... \mathbf{e}_s^1\}$ and the intra-image hyperedges set $E^2 = \{\mathbf{e}_1^2, \mathbf{e}_2^2, ... \mathbf{e}_k^2\}$. An inter-image hyperedge connects local features in two images; we also say it connect two images in this paper. An intra-image hyperedge connects local features in the same image. These hyperedges are shown by the orange and blue groups in Figure 1-b). The hypergraph has two incidence matrixes $\mathbf{S}^1 = (s_{ia}^1)$ and $\mathbf{S}^2 = (s_{ia}^2)$, where $s_{ia}^\alpha$ is 1 if $\mathbf{y}_i \in \mathbf{e}_a^\alpha$ and 0 otherwise. We build the ordinary Knn offline and build the hypergraph on top of the Knn graph online. We call the local features in the hypergraph **activated local features**. We now describe how to constructe the hypegraph.

In order to solve the ambiguity problem of graph propagation, hypergraph propagation propagates only among the spatially matched local features of images. This paper uses RANSAC [12], the most common features matching method in the spatial verification stage of image search [25, 21, 5], to pre-compute the homography matrix [12] between two images offline. We then use these pre-computed homography matrixes to verify if two features in two images are matched when building the hyperedges at query time.

We assume the query image can be represented by local features in $Y$ and will discuss how to handle a new query in Section 4.3. Because a propagation process of a query is progressive, we expand hyperedges on-demand. To judge if an image is related to the query in a sequence of similar images, we only need to see if it has the activated local features. An inter-image hyperedge is then defined as follows:

**Definition 4.1** (Inter-image hyperedge). For two images $\mathbf{x}_u$ and $\mathbf{x}_v$, let $V_u \subset Y$ be the activated local features set of $\mathbf{x}_u$, and $I_v \subset Y$ be the local features set of $\mathbf{x}_v$. The inter-image hyperedge among features in $\mathbf{x}_u$ and $\mathbf{x}_v$ is: $\mathbf{e}^1 = ( In = V_u, OUT = \{ \mathbf{y}_j \in I_v; \exists \mathbf{y}_i \in IN$ is matched with $\mathbf{y}_j$ by the pre-computed homography between $\mathbf{x}_u$ and $\mathbf{x}_v \} )$.

If the matching inliers between two images are less than a threshold, they could be directly verified as irrelevant with each other. As the target of hypergraph is to search the related images for the query, we do not build the hyperedge for them. We set the threshold as 20 in this paper, which is the standard threshold in the spatial verification stage of the image search systems [23, 7]. Note that even if an image has more than 20 matching inliers with another one, its activated local features may not be matched.

Similar to object detection [28], we use bounding box to describe the object region. In an image, local features depicting an object locate in the object's bounding box. As a homography does not guarantee to find all the matched features between two images, the inter-image hyperedges do not connect all the features in an object's bounding box. Unfortunately, the unmatched related features are also important for propagation. As shown in Figure 1-b), features $\mathbf{y}_1$, $\mathbf{y}_2$ and $\mathbf{y}_3$ are in the same region and describing the same building, but only $\mathbf{y}_1$ and $\mathbf{y}_3$ are connected within image $\mathbf{x}_2$ using an inter-image hyperedge to image $\mathbf{x}_1$. Without an intra-image hyperedge connecting $\mathbf{y}_1$, $\mathbf{y}_2$ and $\mathbf{y}_3$, we cannot propagate information from images $\mathbf{x}_1$ to $\mathbf{x}_3$. To recover the unmatched yet related local features, we define the intra-image hypergraph as follows:

**Definition 4.2** (Intra-image hyperedge). Given a feature set in an image $\mathbf{x}_u$, let $V_u \subset Y$ be the activated local features set of $\mathbf{x}_u$, and $I_u \subset Y$ be the set of all the local features of $\mathbf{x}_u$. Let the smallest rectangle covering all the features in $V_u$ be the bounding box $b$. The intra-image hyperedge in $\mathbf{x}_u$ is defined as $\mathbf{e}^2 = ( IN = V_u, OUT = \{\mathbf{y}_j \in I_u; \mathbf{y}_j$ exists in $b\})$.

Due to the high computing cost, it is impractical to perform the propagation on a large-scale dataset consisting of millions of images. Inspired by the idea of truncation [16] in diffusion, we build inter-image hyperedges only among the hop-$N$ neighbors [1] of a given query image. Specifically, we build the inter-image hyperedges from image $\mathbf{x}_u$ to $\mathbf{x}_v$ only if both of them are in the hop-$N$ neighbors of the query.

---

[1] Node A is in B's hop-$N$ neighbors if A can be connected with B using not more than $N$ different hyperedges.

## 4.2 Propagation

After building the hypergraph, we use a propagation model in the hypergraph to achieve the final ranking scores of retrieved images. Specifically, we define a propagation matrix $\mathbf{P} = (p_{ij})$ and set the propagation weight between $\mathbf{y}_i \in \mathbf{x}_u$ and $\mathbf{y}_j \in \mathbf{x}_v$ as $p_{ij} = d(\mathbf{x}_u, \mathbf{x}_v) \max_{abc}(s_{ia}^1 s_{ba}^1 s_{bc}^2 s_{jc}^2)$, where $d(\mathbf{x}_u, \mathbf{x}_v)$ is the Euclidean distance between the global features of $\mathbf{x}_u$ and $\mathbf{x}_v$, and $\max_{abc}(s_{ia}^1 s_{ba}^1 s_{bc}^2 s_{jc}^2)$ records whether $\mathbf{y}_i$ and $\mathbf{y}_j$ are connected in the hypergraph. The propagation matrix is normalized using the degree matrix $\mathbf{D} = diag(\mathbf{P1}_n)$ as $\mathbf{P}' = \mathbf{PD}$. For aggregation function, $a_{iu} \in \mathbf{A}$ is set as $\frac{1}{b_u}$ if $\mathbf{y}_i \in \mathbf{x}_u$, and 0 otherwise, where $b_u$ is the number of activated local features in $\mathbf{x}_u$. Then the propagation function in Equation 2 can be written as:

$$\mathbf{l}^* = I_h(\mathbf{l}^0) = \mathbf{l}^0 + \mathbf{l}^0 \mathbf{P}' + \mathbf{l}^0 \mathbf{P}'^2 + \cdots + \mathbf{l}^0 \mathbf{P}'^N, \mathbf{f}^* = Agg(\mathbf{l}^*) = \mathbf{l}^* \mathbf{A}. \tag{3}$$

Intuitively, this propagation function gives high scores to the images with shorter hypergraph geodesic distances to the query while considering similarities of global features.

## 4.3 Community selection for new query

The hypergraph propagation described in Sec. 4 assumes the query is in the database, which may not be true in a general retrieval scenario. Now we consider how to find the good starting nodes for a new query.

As discussed in Section 3, we observe that the items containing the same object usually belong to the same community in $G$. By checking whether the top S items of the initial search are in the same community, we can evaluate the quality of the initial search without query time spatial verification. As shown in Figure 2, the quality of the global-feature-based search of Q1 is better than Q2 because its top-ranked items are in the same community.

Specifically, after the initial search, we use the top $S$ images and their connected edges in $G$ to build a subgraph $G_s$. $G_s$ consists of one or more components, and each component represents a community in $G$. We call the top-1 image of initial search the dominant image, its component in $G_s$ the dominant component, and its community in $G$ the dominant community. We call other components in $G_s$ as the opposition components. Intuitively if all the top $S$ images in $G_s$ constitute a single component, all of them may contain the same object with the query image. Otherwise, the uncertainty will increase if the number and size of the opposition components increase. This observation also agrees with our experimental results. Inspired by Shannon entropy, we define the uncertainty index, $U$, of the initial search as:

$$U = -\sum_{i=1}^{n_G} P(C_i) \log P(C_i), \tag{4}$$

where $n_G$ is the number of component in $G_s$, $C_i$ is a component and $P(C_i)$ is the proportion of $C_i$ in $G_s$.

Community selection is the pre-stage of hypergraph propagation. The overall retrieval process with community selection is to perform the initial search firstly and then calculate the uncertainty of the initial search result. If the uncertainty index is low, say $U$ is lower than 1, hypergraph propagation uses images in the dominant community to represent the query image and starts the propagation process from them. If the uncertainty index is high, the search engine does the spatial verification on

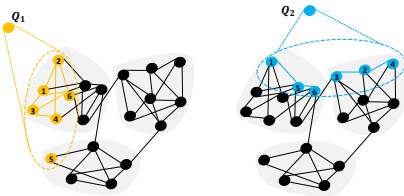

Figure 2: Two queries on an image graph with three communities. The numbers on the nodes are their rankings in the initial search. The uncertainty of the initial search result of Q1 is lower than that of Q2 as most retrieved items of Q1 distribute in the same community.

the top $T$ images of the initial search to find a new dominant community for the query. In this way, we significantly reduce the number of spatial verification while keeping very high accuracy.

# 5 Experiment

## 5.1 Implementation

We compare hypergraph propagation against the existing QE/diffusion methods: average query expansion [9], average query expansion with decay [13], $\alpha$ query expansion and diffusion [11]. We evaluate the performance of hypergraph propagation under the ordinary retrieval pipeline: performing the initial search, followed by conducting QE/diffusion. The focus of hypergraph propagation is on improving the QE/diffusion process, so we do not consider more complex approaches such as repeating QE and diffusion in different stages (e.g., initial search+ QE + spatial verification + Diffusion [6] and ASMK + spatial verification + Diffusion [25]). Fortunately, our system can be combined with other methods and produce stronger performance thanks to its generality. We do not use community selection and always start the hypergraph propagation from the initial dominant community for all the queries to make the comparison fair.

For hypergraph propagation and all the compared QE/diffusion methods, we use the DELG model [5] trained on GLDv2 [35] to extract both global and local features. We find the nearest neighbor number $K$ does not influence the final result a lot, and we report the performance of setting $K$ as 200. After the propagation, we reorder the initial ranking list using $\mathbf{f}^*$ to get the final results.

For community selection, we set $S$, the number of images for calculating the uncertainty, as 20. When testing the performance of combining community selection and hypergraph propagation, we set the uncertainty threshold as 1, which we find is a good balance of the accuracy and computation overhead. For the query with an uncertainty index higher than the threshold, we do the spatial verification for the top 100 items using SIFT features [18]. Once we find an item with more than 20 inliers, we treat its community as the new dominant community, stop the spatial verification, and apply hypergraph propagation in the new dominant community. The code of this work is publicly available on `https://sgvr.kaist.ac.kr/~guoyuan/hypergraph_propagation/`

We evaluate our methods on two well-known landmark retrieval benchmarks: revisited Oxford (ROxf) and revisited Paris (RPar) [25]. There are 4993 (6322) database images in the ROxf (RPar) dataset, and a different query set for each dataset, both with 70 images. We also report the result with the R1M distractor set, which contains 1M images. The accuracy is measured using mean average precision (mAP). We conduct the offline process on 6 Intel(R) Core(TM) i9-9900K CPUs @ 3.60GHz and 896GB of RAM and implement the online hypergraph propagation and community selection on one CPU and 50GB of memory.

## 5.2 Comparison to the state-of-the-art

Table 1 compares the accuracy of hypergraph propagation and other QE/diffusion methods. Our method obtains the best result on ROxford and RParis, both with and without the R1M distractor set. Compared to RParis, ROxford is more challenging as it contains more difficult retrieval cases such as the change of viewpoint [25]. The performance improvement of our method on ROxford hard cases is impressive; we get the mAP 70.3 and 60 with and without 1M distractors, outperforming the traditional diffusion method 18.5% and 55%, respectively. To the best of our knowledge, these are the state-of-the-art mAP even compared with other techniques methods integrating additional steps [25, 6] with more computation and memory requirements.

Ordinary diffusion is better than QE methods on all the datasets except ROxford with R1M distractors. We think that the texture and shape of medieval buildings in ROxford are common and tend to appear in other buildings around the world. When retrieving these objects with R1M distractors, its ambiguity problem explores more false positive items, leading to a low mAP than QE methods. Our hypergraph propagation keeps the spatial information during propagation, thus avoiding the false positive issue of ordinary diffusion, resulting in high accuracy.

Table 1: Results (% mAP) on the ROxf/RPar datasets and their large-scale versions ROxf+1M/RPar+1M, with both Medium and Hard evaluation protocols.

| Method | ROxf M | ROxf H | ROxf+R1M M | ROxf+R1M H | RPar M | RPar H | RPar+R1M M | RPar+R1M H |
|---|---|---|---|---|---|---|---|---|
| DELG (Global) [5] | 76.3 | 55.6 | 63.7 | 37.5 | 86.6 | 72.4 | 70.6 | 46.9 |
| DELG (Global + Local) [5] | 81.2 | 64.0 | 69.1 | 47.5 | 87.2 | 72.8 | 71.5 | 48.7 |
| Average QE [9] | 77.2 | 57.1 | 68.5 | 43.0 | 87.6 | 74.3 | 75.4 | 54.8 |
| Average QE with decay [13] | 78.4 | 58.0 | 70.4 | 44.7 | 88.2 | 75.3 | 76.2 | 56.0 |
| $\alpha$ QE [27] | 65.2 | 43.2 | 57.0 | 30.2 | 91.0 | 81.2 | 81.0 | 64.1 |
| Diffusion [16] | 81.0 | 59.3 | 63.9 | 38.7 | 91.4 | 82.7 | 80.0 | 64.9 |
| Hypergrah Propagation (Ours) | **85.7** | **70.3** | **78.0** | **60.0** | **92.6** | **83.3** | **86.6** | **72.7** |

## 5.3 Effectivness of community selection

Figure 3 shows the mAP of the initial search results in ROxford with different uncertainty thresholds. As the uncertainty increases, the mAP of the initial search result decreases. The uncertainty index predicts the quality of the initial search without any user feedback. Only for the high uncertainty retrieval results, we use the computationally heavy approach, i.e., spatial verification, to improve the precision of nearest neighbors before the hypergraph propagation.

Table 2 compares the performances between with and without community selection (CS) as the pre-stage when using spatial verification (SP) to initialize the hypergraph propagation (HP). With CS, the search engine improves the performance of hypergraph propagation in ROxford very efficiently by only conducting SP on 4 and 7 queries with and without R1M distractors, respectively. The performance improvement with R1M distractors is less than that without them. We think its reason is that the correct items with the R1M distractors are ranked outside the top 100 images and thus are not detected by SP. Although CS does not improve the retrieval results of these cases, it successfully tags them as uncertain cases. We think this result leads to a promising future direction. Because once a search engine can automatically detect the failed query cases, it is possible to refine the representation ability by relabeling and training only for the failed cases instead of training on a large dataset.

Thanks to the improved representation ability of DELG [5] features, the nearest neighbors of all queries in RParis dataset have high precision. The average precisions of the top 20 items of the initial result of all the queries in RParis are 0.97 and 0.90 in medium and hard protocols, respectively. As the initial dominant communities for all the queries in RParis are correct, online SP is not helpful for the propagation process. CS avoids these unnecessary SPs in these cases as the uncertainties of all the queries are low, as shown in Table 2. CS significantly improves the retrieval performance for some hard cases. Figure 4 shows an impressive example in ROxford, in which the mAP increases from 0.077 and 0.062 to 0.988 and 0.987 in medium and hard protocols, respectively.

In summary, the main benefits of CS include: 1) CS quickly detects the low-quality retrieval result, which is either because of the difficult query or the insufficient feature representation ability; 2) CS avoids the unnecessary computationally heavy spatial verification process; 3) CS provides significant performance improvement for some hard cases.

## 6 Discussion

While we have shown benefits of our method, it has limitations and interesting future work. First, matching using the homography matrix calculated by RANSAC might not be the best option for

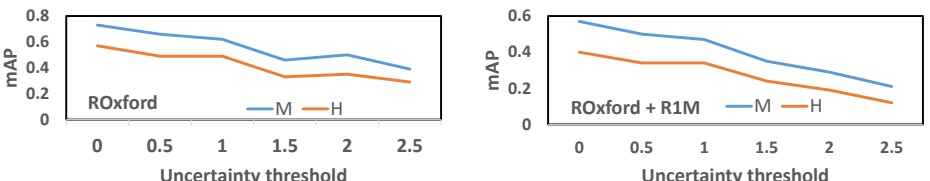

Figure 3: mAP of the query results above the uncertainty threshold. Uncertainty predicts the query quality well.

Table 2: Comparison between with and without community selection (CS) as the pre-stage when using spatial verification (SP) to initialize the hypergraph propagation (HP). This table shows both the mAP and the total number of queries which need SP when testing the corresponding dataset.

| | HP | CS + SP + HP | | SP + HP | |
|---|---|---|---|---|---|
| | mAP | mAP | total # SP | mAP | total # SP |
| ROxf(M/H) | 85.7/70.3 | 88.4/73.0 | **4** | 88.4/73.0 | 70 |
| ROxf+R1M(M/H) | 78.0/60.0 | 79.1/60.5 | **7** | 79.1 / 60.5 | 70 |
| RPar(M/H) | 92.6/83.3 | 92.6/83.3 | **0** | 92.6/83.3 | 70 |
| RPar+R1M(M/H) | 86.6/72.7 | 86.6/72.7 | **0** | 86.6/72.7 | 70 |

our hypergraph propagation. It does not consider local features' contextual cues when applying matching, and it has the offline homography calculation overhead. Recent deep-learning-based matching techniques such as SuperGlue [29] show performance improvement in many tasks in terms of both speed and accuracy and can also be helpful for our hypergraph propagation. Second, to achieve a better result, both hypergraph propagation and community selection can be combined with existing query expansion, diffusion, and spatial verification methods. For example, it is possible to adapt region diffusion [10] to the hypergraph model or adapt community selection on ordinary diffusion methods.

Image search techniques might be related to surveillance techniques. But our work does not study human-related problems, so we think the negative social impact is minor.

Overall, we believe that our hypergraph propagation method combined with the community selection technique takes a meaningful step towards designing highly accurate image search techniques. Especially on the hard protocol of the ROxford dataset, our hypergraph propagation achieved the impressive mAPs of 73.0 and 60.5 with and without R1M distractors, respectively. To the best of our knowledge, these are the state-of-the-art results.

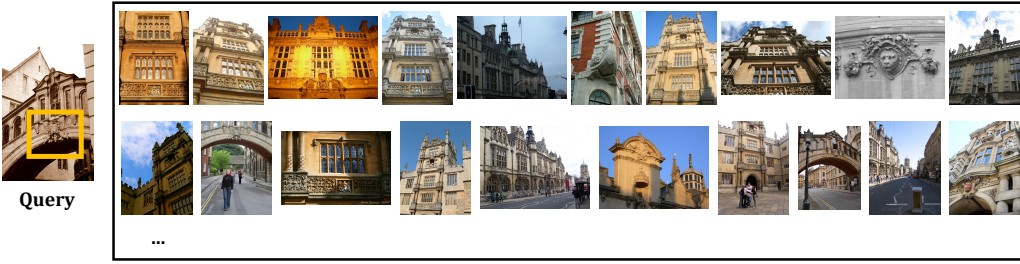

The initial search result ( uncertainty: 1.77, Medium mAP: 0.077, Hard mAP: 0.062 )

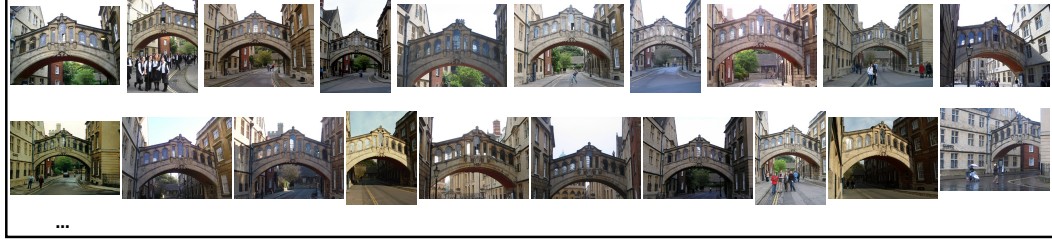

The result after CS and HP ( Medium mAP: 0.988, Hard mAP: 0.987 )

Figure 4: In this example, the top 11 images and the initial dominant community is unrelated to the query. Its uncertainty index is 1.77, which means the search engine is very uncertain about the accuracy of initial dominant community. So CS finds a new dominant community using spatial verification. After hypergraph propagation in this new dominant community, the retrieval performance is significantly improved.

## Acknowledgments and Disclosure of Funding

We would like to thank the anonymous reviewers for their valueable comments and insightful suggestions. Sung-Eui Yoon and Yuchi Huo are co-corresponding authors of the paper. This work was supported by Institute of Information & communications Technology Planning & Evaluation (IITP) grant funded by the Korea government (MSIT) (IITP-2015-0-00199, Proximity computing and its applications to autonomous vehicle, image search, and 3D printing) and the National Research Foundation of Korea (NRF) grant funded by the Korea government(MSIT) (No. 2019R1A2C3002833).

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
