Table 1: The data size and query speed on ROxford + R1M.

(a) Average data size per image. Note that the local descriptors are not required in the online retrieval.

| global feature | local features' matching information | local features |
|---|---|---|
| 8192 B | 2678 B | 1040000 B |

(b) Breakdown of average time per query.

| initial search | hypergraph propagation | uncertainty calculation | spatial verification |
|---|---|---|---|
| 0.62 s | 1.07 s | 0.0003 s | 41.12 s |

# Appendix A: Data size and speed

To construct hypergraph at query time, we need to record the spatial matching information for each image and its k-nearest neighbors. Instead of pre-computing and recording the homography matrices as described in the Section 3, we directly record matched local feature pairs. More specifically, we conduct spatial verification between each database image and its K-nearest neighbors, then record the local features' locations and a set of index tuples pointing to matched local-feature pairs.

Table 1 (a) summarizes the average size of the global feature, local features and the local features' matching information per image when conducting the experiment on ROxford + R1M. Note that the online retrieval only uses the locations of local features, therefore we do not need to load the local features into memory. The size of local features' matching information is less than that of the global feature.

Table 1 (2) shows the breakdown of average time per query during experiment on ROxford + R1M. The average time of hypergraph propagation for each query is 1.07 s while initial search is 0.62 s. We think this overhead is acceptable. In addition, the spatial verification takes much longer time than other parts. This verifies the efficiency of our community selection approach, which fastly calculates the uncertainty to skip the unnecessary spatial verification for good initial search result.

# Appendix B: Visualization of hypergraph propagation

We visualize the benefit of hypergraph propagation with actual queries in Figure **??**. The yellow boxes represent the correctly matched regions through hyperedges. In each row, the first image and the third image are wrongly connected through the second image in the ordinary graph. Our hypergraph propagation mechanism correctly separates these wrong connections by solving the ambiguity problem of propagation.

# Appendix C: Illustration of hypergraph propagation

Appendix-Figure 2 shows a large version of Figure 1 in the paper. In Appendix-Figure 2, a) shows a part of an ordinary graph with scalar-weighted, i.e., similarity, edges. Orange frames are the common visible regions among images $\mathbf{x}_1$, $\mathbf{x}_2$, $\mathbf{x}_3$, and $\mathbf{x}_4$. Purple frames are the common visible regions between images $\mathbf{x}_2$ and $\mathbf{x}_5$. $\mathbf{x}_3$ and $\mathbf{x}_5$ are close neighbors to image $\mathbf{x}_2$. While $\mathbf{x}_3$ is related to $\mathbf{x}_1$ by sharing the orange frame, $\mathbf{x}_5$ is not. Utilizing scalar-weighted edges cannot propagate the query in the ordinary graph without this ambiguity issue. b) shows the corresponding hypergraph of a). Inter-image hyperedges $\mathbf{e}_s^1$ are shown in yellow, intra-image hyperedges $\mathbf{e}_k^2$ are in blue, and local features $\mathbf{y}_n$ are in green. A hypergraph path connects local features from $\mathbf{y}_1$ to $\mathbf{y}_9$ in $\mathbf{x}_1$ and $\mathbf{x}_3$, but no path connects local features in $\mathbf{x}_1$ and $\mathbf{x}_5$.

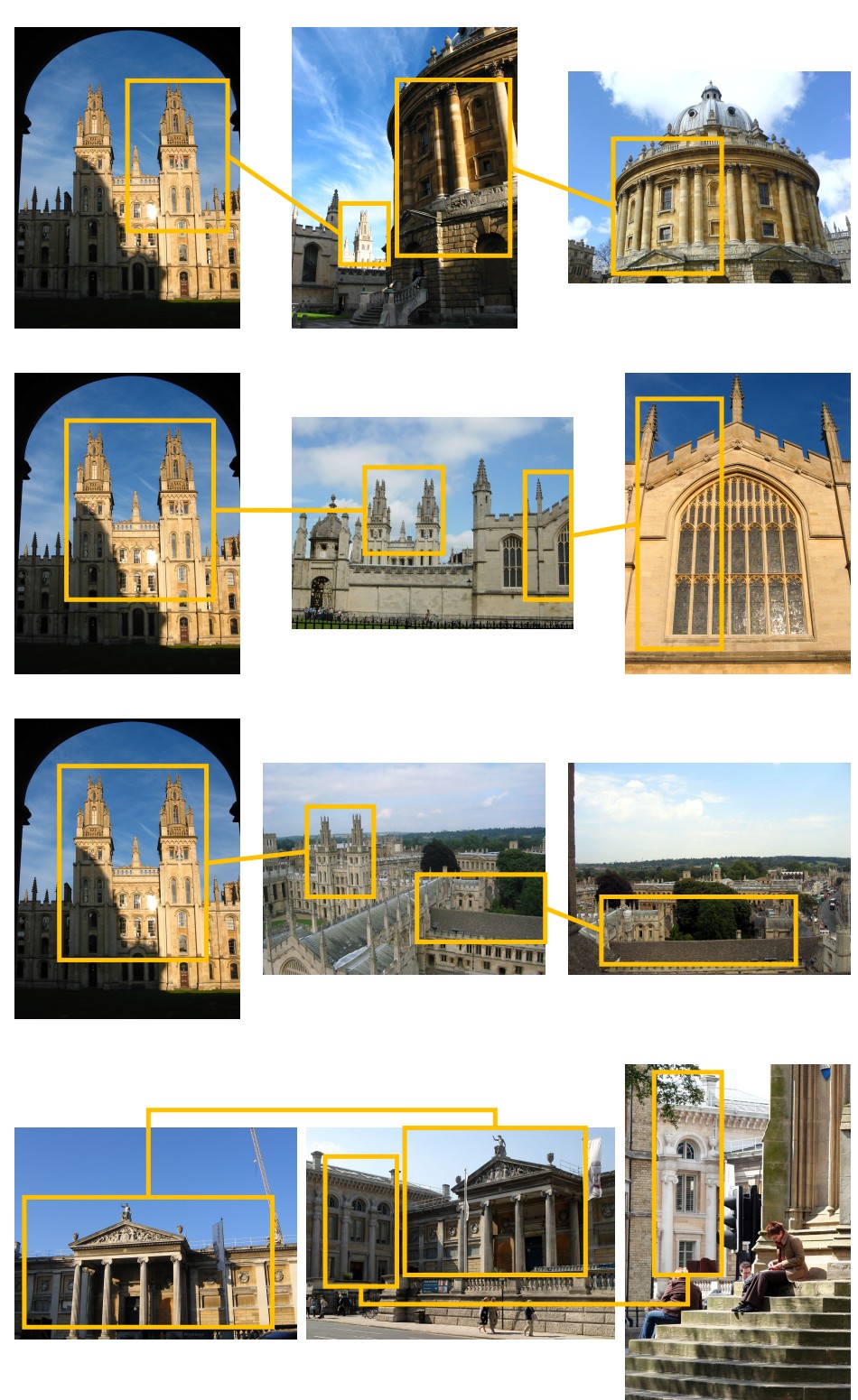

Appendix-Figure 1: Visualization of the benefit of hypergraph propagation.

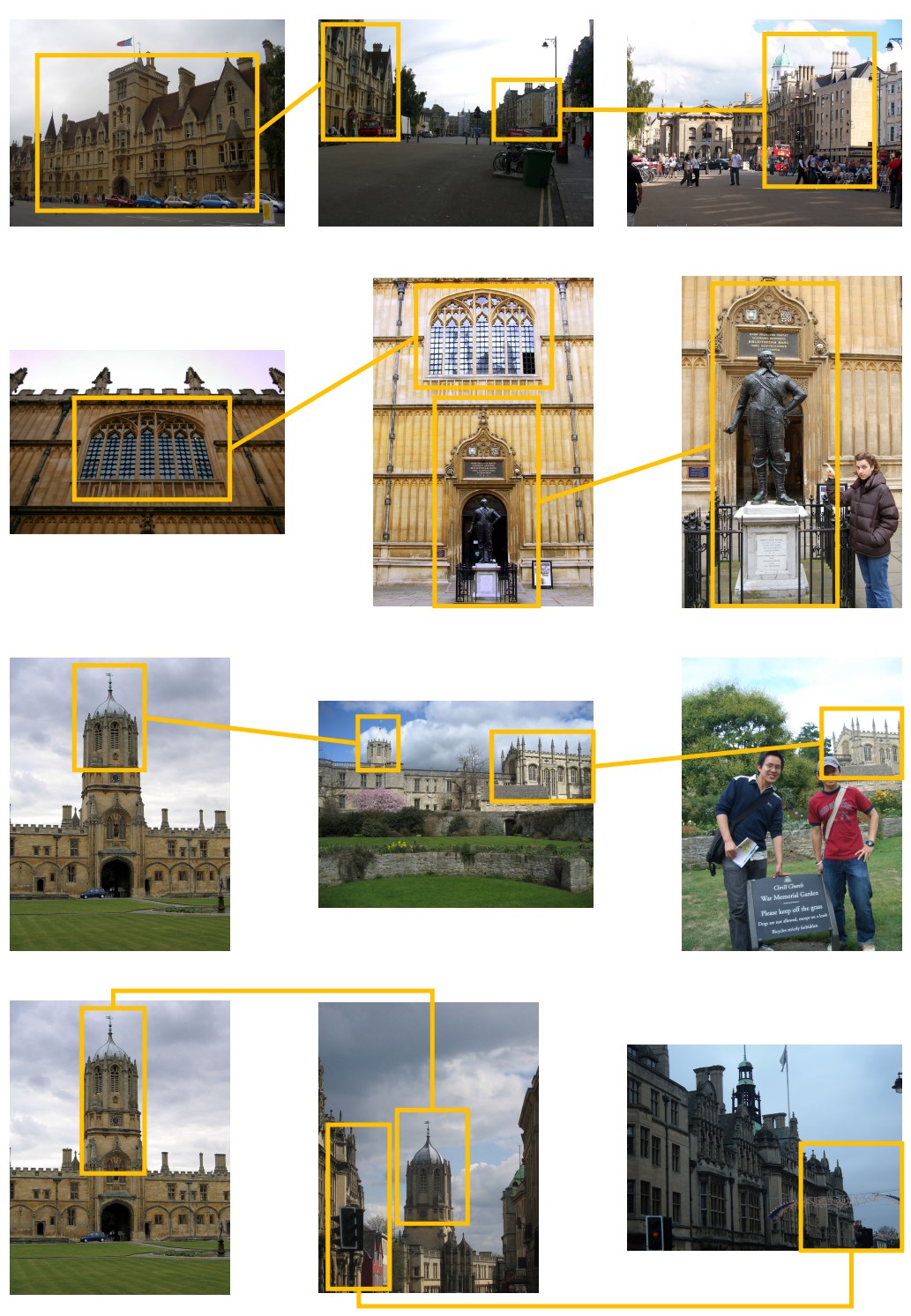

Appendix-Figure 1: Visualization of the benefit of hypergraph propagation.

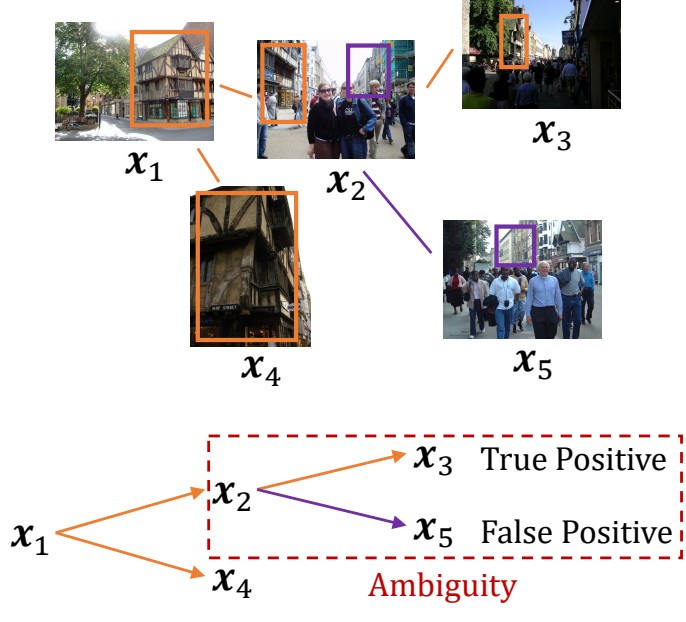

(a) Ordinary Graph Propagation

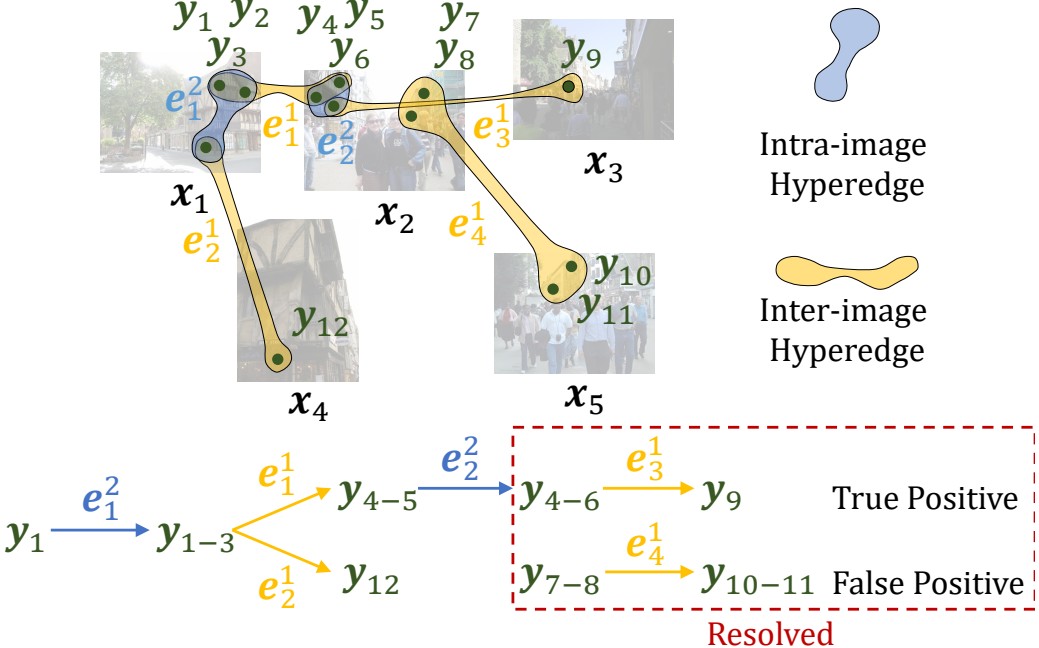

(b) Hypergraph Propagation

Appendix-Figure 2: The large version of Figure 1 in the paper.