# OpenReview forum: "Hypergraph Propagation and Community Selection for Objects Retrieval"
_NeurIPS.cc/2021/Conference — NeurIPS 2021 Poster_

### Official Review · Reviewer_YeE2 · 2021-07-09

**Rating:** 6
**Confidence:** 3

**Summary:**

This paper proposes a new query expansion method for image object retrieval. Firstly, they propose a hypergraph among local features of base images. To expand new queries, it is time-consuming to check whether the query is related to initial search results. They propose a community selection method to speed up the procedure. In experiments, the proposed method outperforms traditional methods.


**Limitations And Societal Impact:**

Yes.

**Main Review:**

Positive points:
The proposed community selection method seems reasonable and is efficient in new query expansion.

My concerns and questions:
1. The motivation for using hypergraphs is weird. At least the example in Figure 1 is misleading. In Figure 1 (a), the ordinary graph is constructed at the image level, so it leads to ambiguity. Can the ambiguity be solved by constructing ordinary graphs by local features as in Figure 1 (b)? Say, x_2 and x_5 are connected based on local features in purple frames while x_2 and x_3 are connected based on orange frames. In this way, the ambiguity would be solved.
2. Inter hyperedges are only constructed between two images. Is it possible to construct a hyperedge involves more than two images?
3. Section 4.1 is difficult to follow. What is the difference between "activated local features" and "local features"? What are "IN" and "OUT"? If they are concepts from other papers. They should be introduced here to let this paper be self-contained. These concepts are important for understanding.

Minor problems:
1. On my side, Figure 1 (b) is difficult to read after printing.
2. Typo: Line 36, 'research' -> 'search'.

**Time Spent Reviewing:**

3

---

> ### Author Response · Authors · 2021-08-10
> **Authors' response to reviewer YeE2**
>
> We appreciate the detailed reviews. We are encouraged that you think our proposed community selection method is reasonable and efficient. We would like to answer your concerns.
>
> ### Q1. The motivation for using hypergraphs is weird. At least the example in Figure 1 is misleading. In Figure 1 (a), the ordinary graph is constructed at the image level, so it leads to ambiguity. Can the ambiguity be solved by constructing ordinary graphs by local features as in Figure 1 (b)? Say, $x_2$ and $x_5$ are connected based on local features in purple frames while $x_2$ and $x_3$ are connected based on orange frames. In this way, the ambiguity would be solved.
>
> Sorry for the confusion and thanks for your good question. Reviewer fAs6 has a similar concern with you. If you want to see a concise answer, you may read our response to question 1 of reviewer fAs6. Based on your question, we want to discuss the following alternative ordinary graph models to clarify the motivation for using hypergraph.
>
> 1) Treat every image as a node, and pre-compute the matched regions between the images connected by the kNN graph, i.e., we only know that $x_2$ and $x_3$ contain the same object in orange frames, but do not know the one to one matching information of local features in the purple frames. We think this approach is sufficient for the easy retrieval task but does not satisfy the generalized object retrieval target. In the real world, an image usually contains many objects, and we could not define what is an object. The user's query could be a street, a building in the street, the roof of the building, or only a small sign on the roof. Say image A contains a building with a roof, and image B contains the same building without its roof. If the query is the roof of the building instead of the building itself, this approach could not answer if B is related to the query without doing geometric verification between them.
>
> 2) Says we treat every local feature as a node but do not build the hypergraph. We not only pre-compute the matched frames, but also the matching information of local features in the matched frames. We connect the matched features using pairwise edges to form an ordinary graph and apply propagation on this ordinary graph directly when the query comes.
> This approach seems good. However, there are up to 1000 local features in every image and currently, no matching algorithm could accurately find all the matching features between two images. As we discussed in Section 4.1 (the paragraph before Definition 4.2), the unmatched related features are also important for propagation. Without the intra-image hyperedge connecting the unmatched related features, the propagation could break down between x2 and x3. As we could not define the queried object in front, the intra-image hyperedge could only be built dynamically after users input the query.
>
>
> 3) Our approach pre-computes the one-to-one matching information of local features, dynamically builds a hypergraph when the query comes, and does the propagation on the hypergraph. We find this approach significantly improves the retrieval performance and achieves the SOTA result on ROxford, as shown in Section 5. Now, one may think that the hypergraph could reduct to pairwise interactions. Could we achieve the same goal by only using an ordinary graph? Why do we have to use hypergraph? Firstly, if we reduct the hypergraph to an ordinary graph with pairwise interactions, the number of edges will increase in order. So propagation on the ordinary graph is inefficient. Secondly, hypergraph has different underlying meanings compared with ordinary graphs, therefore reducing to pairwise interactions might lead to degrading performance during propagation. For example, in Figure Response-1, we can not distinguish case (a) and case (b) if we replace the intra-image hyperedges with the pairwise edges. Lastly, using hypergraph is more intuitive by considering the construction and propagation process, wherein every construction step we explore a group of local features from another group of local features.
>
> ### Q2. Inter-image hyperedges are only constructed between two images. Is it possible to construct a hyperedge involves more than two images?
>
> It is a good point. As we discussed in question 1, the principle of designing the hypergraph in this work is to solve the practical problems of the image retrieval engine. We claim the proposed model nicely combines the local and global features and achieves impressive performance improvement. There might be some variants about how to build the hypergraph, and we believe the idea you mentioned is interesting. Constructing a hyperedge involving more than two images may make our work neater and make the propagation efficient. However, it may also bring in some technical problems. For example, if one image contains a building, one image contains the roof of the building, and one image contains the window of the building, should we connect these three images using one hyperedge? We could not give a detailed comparison between the multi-image hyperedge and bi-image hyperedge now. We think this is an interesting topic to explore in the future.
>
> ### Q3. Section 4.1 is difficult to follow. What is the difference between "activated local features" and "local features"? What are "IN" and "OUT"? If they are concepts from other papers. They should be introduced here to let this paper be self-contained. These concepts are important for understanding.
>
> Sorry for the confusion. We would like to give more explanation about 'activated local features', 'IN' and 'OUT' here.
>
> 'local features': a general word. Every image is extracted up to 1000 local features.
>
> 'activated local features': If a local feature is included as a node in the hypergraph, we call it is activated.
>
> 'IN' and 'OUT': They represent the tail and head of a hyperedge. The hyperedge in a directed hypergraph is usually defined as an ordered pair of node sets, constituting the tail and head of the hyperedge, such as " (TAIL = {y1,y2}, HEAD = {y3,y4}) ". In this paper, we use 'IN' and 'OUT' to replace 'TAIL' and 'HEAD'.
>
> Because Section 4.1 is long, we can not interpret all the concepts here. But we will add more explanations of the important concepts, refer to the papers using similar concepts and highlight the keywords to make Section 4.1 easier to follow.
>
> ### Minor problems.
> ### 1. On my side, Figure 1 (b) is difficult to read after printing.
> ### 2. Typo: Line 36, 'research' -> 'search'.
>
> Thanks. For Figure 1, we are thinking about giving a concise version or give a large-size version in the Appendix. And we will check this paper again to modify the typo errors.
>
> Thanks for your precious comments again.

---

> > ### Author Response · Authors · 2021-08-11
> > **Figure Response-1.**
> >
> > Please check Figure Response-1 from: https://imgur.com/a/KwpTFyD
> >
> > Figure Response-1. We can not distinguish case (a) and case (b) if we replace the intra-image hyperedges with the pairwise edges.

---

### Official Review · Reviewer_PYXJ · 2021-07-14

**Rating:** 8
**Confidence:** 5

**Summary:**

This work tackles the problem of image retrieval, specifically in the domain of re-ranking with ambiguous objects. It proposes two core ideas to improve upon existing query expansion and diffusion techniques, namely

1) Introduction of the hypergraph which unifies both global and local relations in the graph propagation process, hence resolving potential ambiguities.
2) Community selection, which resolves the ambiguities in initializing the graphs.

The proposed method shows strong performance on the standard ROxf-RPar (and the 1M distractor set) datasets over existing image retrieval methods with the re-ranking process. Qualitative results are also provided to help understand the way it increases the precision of top-ranked results.

**Limitations And Societal Impact:**

The authors have addressed the limitations and potential negative societal impact of this work.

**Main Review:**

## Originality
To my best knowledge, the concept of hypergraphs for re-ranking is novel to the image search problem. In fact, this work is also the first attempt to combine the two existing classes of re-ranking methods: local descriptor-based geometric verification and global descriptor-based (query expansion and diffusion) into a single framework with the proposal of the hypergraph. I think this is a very elegant way of solving the existing ambiguities mentioned by the authors and the results speak for themselves.

The other novelty proposed in the work, Community selection, is less technical. However, this does not prevent it from being a good novelty as the motivation for it is intuitively explained and has the potential to be adopted in all general image retrieval pipelines.

## Quality
The submission is technically sound.

After assessing the methods and experiments sections, I think the contribution of both novelties is well-justified by the experimental results and ablation studies. It is truly impressive the addition of hypergraph propagation alone already outperforms state-of-the-art (SoTA) in the ROxf-RPar [25] benchmark by a huge margin (12.5% increase in mAP over DELG [5] with local re-ranking on ROxf+1M-Hard, which is notoriously difficult due to the queries having cropped regions that cause extreme zooms *v.s.* database images, a lot of ambiguous regions/textures/objects, let alone the 1-million distractors). The impact of community selection is also clearly justified by the results in Table 2, and the great addition of Figure 4 to qualitatively show how the proposed re-ranking components significantly improve the retrieval results on arguably the most difficult query among the whole dataset.

There are a few minor questions, which clarifications from the author would be greatly appreciated:
1) It is hypothesized in L237-238 that “if all the top $S$ images in $G_s$ constitute a single component, we are very certain that all of them contain the same object with the query image”. Although this is intuitively correct, I wonder if the authors had a sanity check of whether this is actually true, *e.g.* by actually detecting and classifying the objects and checking whether they all fall into the same “community”? Although this does not affect the validity of the contribution of community selection, I think these results would be very interesting for the image retrieval community and provide domain knowledge for future works.
2) Why are the results from [14] not included in Table 1, since they are a very recent work? It would also be interesting to see if the hypergraph propagation can be incorporated with a Transformer (just out of curiosity, not asking for extra experiments).
3) In L277-278 the method runs on multiple CPUs and requires a huge amount of memory (896GB).

    a) Would the performance drop a lot if you limit the RAM usage to a more normal amount (*e.g.* <50GB)?

    b) Could the authors provide a runtime breakdown analysis, especially how long it takes to run on the 1M datasets?

    c) Is GPU acceleration an option? If so, how much does it improve the runtime?

## Clarity
The paper is very well-written, easy to follow. I believe the technical details presented allow for reproduction of the reported results.

## Significance
With reference to the **Quality** section, I believe the contributions in the work would be impactful for the field of image/object search & retrieval, not only by advancing the state-of-the-art by a big step, but also potentially inspire a lot of future work along this line. Clarifications from the authors on 3c) above in **Quality** would be a bonus, as it helps understand whether the compute requirement would be too much for other researchers to replicate and build upon the impressive results reported in this work.


**Time Spent Reviewing:**

6

---

> ### Author Response · Authors · 2021-08-10
> **Authors' response to reviewer PYXJ**
>
> Thanks for your positive feedback and inspiring discussion. We are encouraged that you think our work would be impactful for the field of image search, not only by advancing the state-of-the-art by a big step but also potentially inspire a lot of future works along this line! We think your questions are very insightful thus want to give some discussions here.
>
> ### Q1. It is hypothesized in L237-238 that “if all the top $S$ images in $G_s$ constitute a single component, we are very certain that all of them contain the same object with the query image”. Although this is intuitively correct, I wonder if the authors had a sanity check of whether this is actually true, e.g. by actually detecting and classifying the objects and checking whether they all fall into the same “community”? Although this does not affect the validity of the contribution of community selection, I think these results would be very interesting for the image retrieval community and provide domain knowledge for future works.
>
> We visualize the kNN graph of ROxford and RParis and observe the location of the positive images of each query. We find that the 'easy' positives of a query always appear in the same community, while the 'hard' positives may not. However, these are only some observations. Thanks to your comment, we realize that the sentence "if all the top $S$ images in $G_s$ constitute a single component, we are very certain that all of them contain the same object with the query image" is not rigorous. We will tone down this sentence as "Intuitively if all the top $S$ images in $G_s$ constitute a single component, it is highly possible all of them contain the same object with the query image. This observation also agrees with our experimental results."
>
> ### Q2. Why are the results from [14] not included in Table 1, since they are a very recent work? It would also be interesting to see if the hypergraph propagation can be incorporated with a Transformer (just out of curiosity, not asking for extra experiments).
>
> We also noticed that [14] is a very promising work, which smartly combined transformer and query expansion. However, the project is released quite recently before the NeuIPS deadline and we do not have enough time to retrain the model of [14] to conduct a fair comparison on the same features. We are glad to report the results of [14] in the future when the experiment is finished.
>
> We also think incorporating hypergraph propagation and transformer is a very interesting direction. For example, on the basis of [14], we may train the aggregator based on the local features. Alternatively, we can add the local features' location in the image and their geometric distances with the query in the hypergraph model to the positional encoding of the transformer.
>
> ### Q3. In L277-278 the method runs on multiple CPUs and requires a huge amount of memory (896 GB).
>
> Sorry for the confusion. The multiple CPUs and a huge amount of memory are only for speeding up the offline process. For the online hypergraph propagation and community selection, one CPU and 50GB is enough.
>
> ### Q3-a). Would the performance drop a lot if you limit the RAM usage to a more normal amount (e.g. $<$ 50GB)?
>
> No, the performance will not drop if we limit the RAM usage to a normal amount. To accelerate the speed of offline geometric verification, we load the local features of distractors to RAM. This is why we use a huge amount of memory. However, the large RAM is not necessary. One can also divide all the geometric verification pairs into batches, and only load the corresponding local features when dealing with every batch. The performance will not get influence.
>
> In the online process, hypergraph propagation only requires additional memory less than 3KB's matching information for each database image, as shown in Appendix A. It also does not require the local features during the query time. So a 50GB is also enough for the online large-scale hypergraph propagation.
>
> ### Q3-b). Could the authors provide a runtime breakdown analysis, especially how long it takes to run on the 1M datasets?
>
> Yes. We offered the runtime breakdown of ROxford + R1M in Table 1 - b) in Appendix. The average time of hypergraph propagation for each query is 1.07 s while an initial search is 0.62 s. We think this overhead is acceptable.
>
> ### Q3-c). Is GPU acceleration an option? If so, how much does it improve the runtime?
>
> Yes, GPU will significantly accelerate the offline geometric verification process. During the offline geometric verification, every pair is independent of the others. So the time complexity is $O ( 1 )$. We open 8 threads on every CPU to simultaneously do the geometric verification. If GPU allows us to open thousands of threads, we theoretically can get a hundred times' speed up.
>
> For the online process, as discussed in question 2 b), we implement the hypergraph propagation using python and observe a reasonable overhead that is comparable with initial research. Therefore we do not explore GPU or C language-based acceleration in our current implementation. However, the computational-heavy part of hypergraph propagation, such as calculating scores, can be handily accelerated by GPUs for production-ready deployment.

---

### Official Review · Reviewer_fAs6 · 2021-07-17

**Rating:** 5
**Confidence:** 3

**Summary:**

This paper proposes a hypergraph model to settle the ambiguity problem of propagation.

**Limitations And Societal Impact:**

1, In the introduction section of the story's beginning, there is no sufficient instruction on why we have to use a hypergraph. What are the benefits of the hypergraph?

2, According Fig. 1 a), the ordinary graph is not the key factor to link false-positive images, since we could also link the similar features in local ranges to break the connection between x1 and x5.

3，The propagation method is still quite simple based on Euclidean distance and constrained by hyperedges. There is not an explicit difference between the proposed hypergraph and an ordinary graph.

**Main Review:**

originality, from technical perspectives, the proposed model is quite simple and common. No many brilliant designs are shown.
quality, it outperforms the listed state-of-the-art methods.
clarity, in the introduction, there is no sufficient instruction to make me understand the necessity of using the hypergraphs.
significance, there are novel designs but this work also has some problems.

**Time Spent Reviewing:**

1 day

---

> ### Author Response · Authors · 2021-08-10
> **Authors' response to reviewer fAs6**
>
> Thanks for your thoughtful feedback, and we are glad that you think we have novel designs. We would like to answer your concerns here.
>
> ### Q2. Why do not directly apply local descriptor-based geometric verification between x1 and x5 to break their connection?
>
> The reason is the limited computing resource.
>
> 1) Notice that we have millions of images in the database and a square of millions of image pairs. It is impractical to do the geometric verification on all the pairs, even offline.
>
>  2) In practice, image search engines usually build the kNN graph instead of the complete graph, which means every image only connects with its top k similar images [16]. Unfortunately, x1 and x5 are not similar at all; there is no reason to do the geometric verification for them in advance. You may want to check more examples in appendix B.
>
>  3) Our main contribution is to answer whether the search engines can know the spatial matching relations between two images without directly applying geometric verification between them. This approach is novel from existing post-processing techniques, which consider the propagation and matching problem separately. Actually, this is the first attempt to combine the two existing classes of re-ranking methods: local descriptor-based geometric verification and global descriptor-based query expansion/diffusion, as reviewer PYXJ says.
>
> ### Q1. Why do we use hypergraph?
>
> Sorry for the insufficient explanations. To answer this question, we want to discuss two difficulties of object retrieval.
>
> 1) It is hard to define what is an object in advance. The input query could be a street, a building in the street, the roof of the building, or only a small sign in the roof. We never know what users want to search before they really input the query. Simply telling two images in the ordinary graph contain the same street or same building is not enough for the propagation of unknown queries; we want to find all their matched patterns.
>
> 2) Modern deep learning based feature extractors extract up to 1000 local features for every database image, it is non-trivial and impractical to directly apply diffusion on the local features given the huge amount of local features. Our approach is to use the hyperedges to connect the matched local features, where the propagation can be performed efficiently. It also gives consideration to both local features and global features to get a better retrieval result. We discussed the ordinary options in the response to reviewer 4.
>
> ### Q3. The propagation method is still simple based on Euclidean distance and constrained by hyperedges. There is not an explicit difference between the proposed hypergraph and an ordinary graph.
> In this work, we focus on improving image search performance and do not claim the contribution of a novel propagation function in graph mining. The propagation function on hypergraph is inspired by the ordinary graph, but the performance, function, structure, using information and the motivation between the proposed hypergraph and the common ordinary graph in image search is quite different. Actually, we deliberately use the basic propagation function to show that the proposed hypergraph model itself significantly improves the retrieval performance, as shown in Table 1 of the paper.
>
> Traditional ordinary-graph-based methods, such as query expansion and diffusion, aim to settle the manifold problem in image search and use only the global features. It is known that diffusion adopts false positive items in practice, and how to settle the false positive is an open problem. In this work, we propose the hypergraph model to elegantly link the spatial verification and query expansion/diffusion to settle this problem. Through both quantitative and qualitative results, we show that our method gives promising performance improvement without high additional computing costs. Our comparison between the hypergraph model and other ordinary graph methods in the response to Reviewer YeE2 also clarifies the benefits of the proposed hypergraph.
>
> Thanks to your comments; we realized the confusing points of the submitted manuscript, especially about Figure 1 (Q2) and the motivation of the hypergraph (Q1). We will add more introduction about them.

---

### Official Review · Reviewer_2ivT · 2021-07-20

**Rating:** 6
**Confidence:** 3

**Summary:**

The paper addresses the object retrieval problem from images. Traditional query expansion methods can only utilize very similar images, resulting in low recall. On the other hand, diffusion propagation methods on a image graph often ambiguity problem of propagation. Instead, spatial verification has to be used for the accurate detection of true positive images. However, in an ordinary graph with scalar edge weights, spatial verification cannot be done. Furthermore, spatial verification is very computationally heavy. To address the problem, a hypergraph is proposed to map image's internal structural information, and a community selection technique is employed for hypergraph propagation without heavy spatial verification. Experiments are conducted which show promising results.

**Limitations And Societal Impact:**

fine with me.



**Main Review:**

Strong points:
- The solution is well motivated, intutive and reasonable.
- The paper is well presented and easy to follow, even for someone not specifically working on object retrieval
- Experiments show that the solution is promising, with both quantitative and qualitative results.

Weak points:
- The solution appears straightforward and quite heuristic. Though technically valid, it is not sure if there is any guarantee on the community based method.
- The community method: only top 1 image is treated as the dominant community. There could be a robustness problem if top 1 image happens to be a false positive.
- I did not figure out how the communities are found. Different community detection method may have an impact on the retrieval performance.

-- update after rebuttal --
Based on the rebuttal, I will maintain my current rating.

**Time Spent Reviewing:**

4

---

> ### Author Response · Authors · 2021-08-10
> **Authors' response to reviewer 2ivT**
>
> Thanks for your good comments. We are glad that you think our work is well-motivated, intuitive, reasonable, and promising. We want to recap this work and then give responses to your concerns.
>
> ### Recap 1:
> This work firstly uses **community selection** to find the starting node for a new query. After that, we use **hypergraph propagation** to find the images related to the query.
>
> ### Recap 2:
> Community **selection** is different from community **detection**. Community detection aims to cluster nodes in the graph as communities. Community selection is to answer "whether we can select a community for the initially retrieved nodes?". In other words, it is to evaluate the probability that the initially retrieved nodes are coming from the same community? The output of community selection is the uncertainty value, which guides the search engine on whether to use spatial verification before propagation.
>
> ### Q2. There could be a robustness problem if the top 1 image happens to be a false positive.
> Sorry for not explaining properly in the paper. The proposed community selection is to settle this robustness problem. As we discussed in Section 3 and Section 4.3, we observe that images containing the same object usually belong to the same community. We observed that a query usually falls into either the interior of a community or the boundary of multiple communities. The latter case indicates a hard inter-class query. For example, if the top 1 image happens to be a false positive and the top 2-20 images are true positive, then the search engine will know that the top 1 is from a different community with the top 2-20. It can directly abandon the false positive top 1. If the top 1 image is false positive and the top 2-20 are also false positive, then it is highly likely that they are coming from several different communities. The community selection algorithm finds it is difficult to select the same community for the top 20 images and outputs a high uncertainty value to push the search engine to start the spatial verification to find the true positive. In theory, there is an extreme, but not a zero-probability case that the top 20 are all false positive and belong to the same community. However, we do not observe such an extreme case yet in the studied datasets. Besides, such an extreme case indicates a systematic failure involving the global feature extractors, which might not be resolved only using spatial verification and propagation-based approaches.
>
> ### Q3. How are the communities found? Different community detection methods may have an impact on retrieval performance.
>
> In community selection, we only evaluate whether the retrieved nodes are from the same community by checking if they are connected in the graph. We do not need to use community detection methods to find the whole community.
>
> However, we also think applying community detection methods to our proposed hypergraph model is a promising direction and may mine a lot of interesting knowledge. For example, by applying Louvain or BigCLAM on the image graph, we may find what kinds of objects people like to take photos of, which part of a building the users usually focus on, and which kind of angles and compositions users like to choose. This information will help us manage the image database better. By detecting the hypergraph motif and global structure pattern of the image hypergraph, we can know which vision patterns always appear together and how they interact with each other.
>
> ### Q1. Though technically valid, it is not sure if there is any guarantee on the community-based method.
> To improve the robustness, existing methods do the spatial verification on every retrieved image to check if they are true positive. As the spatial verification is heavy, an important question is whether the graph information and community-based method could help the search engine evaluate the retrieval quality without the heavy spatial verification.
> To the best of our knowledge, this is the first work trying to answer this question. Given that it is difficult to perform mathematical analysis and guarantee on the communities of the real-world datasets in this initial stage, it is a heuristic approach so far. However, we believe that our results on the famous benchmarks give this question a promising answer.
>
> Hope our response could settle your concerns. Thanks for your precious time and comments again.

---

### Decision · Program_Chairs · 2021-09-27

**Decision:**

Accept (Poster)

**Comment:**

This seems like a good approach which reviewers considered well motivated, and with promising results. Some weak points were identified such as lack of theoretical guarantees, and minor weaknesses/lack of clarity in the presentation. Reviewers largely appreciated the author response, and it seems if the changes are made the manuscript can be of good value to the NeurIPS audience.